# Effects of orally administered crofelemer on the incidence and severity of neratinib-induced diarrhea in female dogs

**Michael Guy** [1]*, **Andre Teixeira**[1], **Allison Shrier**[1], **Carol Meschter**[2], **James Bolognese**[3], **Pravin Chaturvedi**[1]

1 Napo Pharmaceuticals, Inc., San Francisco, CA, United States of America, 2 Comparative Biosciences, Inc., Sunnyvale, CA, United States of America, 3 Cytel, Waltham, MA, United States of America

* mguy@jaguar.health

**Data Availability Statement:** All relevant data are within the manuscript and its Supporting Information files.

## Abstract

Targeted therapies have increased cancer therapy-related diarrhea (CTD) burden, with high incidence and/or severity of diarrhea for some agents that inhibit epidermal growth factor receptor and receptor tyrosine kinases. Neratinib is a pan-HER tyrosine kinase inhibitor approved for breast cancer treatment and causes severe diarrhea in >95% of patients. Crofelemer, a novel intestinal chloride ion channel modulator, is an approved antidiarrheal drug for symptomatic relief of noninfectious diarrhea in patients with HIV/AIDS receiving antiretroviral therapy. The objective of this study was to evaluate the effectiveness of crofelemer prophylaxis in reducing the incidence /severity of neratinib-induced diarrhea without concomitant administration of loperamide in female beagle dogs. A pilot study using 3 dogs determined a maximum daily tolerated dose of neratinib was between 40 and 80 mg; this dose would induce a consistent incidence/severity of diarrhea without risking severe dehydration. In the definitive study, 24 female beagle dogs (8/group) received neratinib once daily and placebo capsules (CTR) four times/day, or neratinib once daily and crofelemer 125 mg delayed-release tablets given two times (BID), or neratinib once daily and crofelemer 125 mg delayed-release tablets given four times per day (QID). Fecal scores were collected twice daily using an established canine stool scoring scale called the Purina Fecal Scoring (PFS) System. After 28 days, using analysis of covariance (ANCOVA), dogs in the CTR group had a significantly higher average number of weekly loose/watery stools (PFS of 6 or 7) when compared to either crofelemer BID (8.71±2.2 vs. 5.96±2.2, p = 0.028) or crofelemer QID (8.70±2.2 vs. 5.74±2.2, p = 0.022) treatment groups. The average number of weekly loose/watery stools were not different between the crofelemer BID and QID treatment groups (p = 0.84). This study showed that crofelemer prophylaxis reduced the incidence/severity of neratinib-associated diarrhea in female beagle dogs without the need for any loperamide administration.

**Funding:** The authors received no specific funding for this work.

**Competing interests:** The authors have declared that no competing interests exist.

## Introduction

Cancer therapy-related diarrhea (CTD) is a debilitating side effect in patients receiving targeted therapy, with or without chemotherapy. CTD leads to dehydration, electrolyte and fluid imbalances, renal insufficiency, malnutrition, fatigue, and other issues. CTD severely compromises cancer treatment because it may require dosing holidays and/or dose reductions of cancer drugs [1], resulting in ineffective cancer treatments or, worse yet, resistance and treatment failures. Furthermore, healthcare resource utilization costs are significant [2, 3] for cancer patients with diarrhea. CTD occurs in 50–80% of patients depending on the targeted therapy and/or chemotherapy regimens, and the advent of newer targeted therapies has increased the burden of gastrointestinal toxicity of these drugs because of their mechanism of action, specifically for those agents that target epidermal growth factor receptor (EGFR), tyrosine kinases, and human epidermal growth factor receptor 2 (HER2) [4]. The inhibition of EGFRs by TKIs and other targeted therapies does not alter intestinal barrier integrity. However, due to the inflammation induced by targeted therapies such as neratinib from inhibition of EGFR targets, intestinal chloride ion secretion is increased by inhibition of EGFRs, as EGFRs is a negative regulator of intestinal chloride ion channels [5]. Neratinib causes severe diarrhea as evidenced by a very high incidence in >95% of breast cancer patients receiving the drug [6]. The management of neratinib-induced diarrhea to ameliorate or mitigate the incidence and severity of diarrhea is critically needed.

Crofelemer is a novel antisecretory, antidiarrheal, oral botanical drug purified from the crude plant latex of *Croton lechleri*. Crofelemer elicits its effects on apical membrane transport and signaling processes involved in intestinal chloride ion transport. Crofelemer reduces chloride ion secretion by the cystic fibrosis transmembrane conductance regulator (CFTR) channel through use- dependent inhibitory modulation of CFTR, resulting in a 50% Inhibitory Concentration (IC50) of 6–7 μM [7]. Crofelemer's action resisted washout, with inhibition lasting several hours after washout. Crofelemer was also found to strongly inhibit the intestinal calcium-activated chloride channel TMEM16A, also known as ANO1 and DOG (anoctamin 1 and Discover On Gastrointestinal Stromal Tumors) by a voltage-independent inhibition mechanism, with a maximum inhibition of 90% and an IC50 of 6.5 μM [7]. The dual inhibitory action of crofelemer on two structurally unrelated intestinal chloride ion channels results in its unique and novel physiological antisecretory antidiarrheal effects in reducing fluid and electrolyte accumulation in the gastrointestinal lumen and improving stool consistency.

Crofelemer has been approved by the US FDA based on its safety and efficacy profile for symptomatic relief of chronic noninfectious diarrhea in HIV/AIDS patients receiving antiretroviral therapy [8]. Crofelemer was also evaluated in a prophylactic setting in a randomized open-label phase 2 study for the prevention of CTD in patients with HER2-positive breast cancer receiving HER2 targeted therapy with trastuzumab, pertuzumab, and a taxane with/without carboplatin (HALT-D study) for two cycles [9]. Crofelemer is currently being evaluated in an ongoing phase 3 clinical trial for the prophylaxis of diarrhea in adult patients with solid tumors receiving targeted therapy with or without chemotherapy (NCT04538625; OnTarget study).

Crofelemer has also been studied in patients with diarrhea-predominant irritable bowel syndrome (d-IBS) [10, 11] and Traveler's diarrhea [12] as well as acute infectious diarrhea with pathogens like Escherichia coli [13] and *Vibrio cholerae* [14]. When compared with placebo, crofelemer significantly reduced the frequency of loose/watery stools as well as improved in stool consistency in these patient populations [15–18].

Neratinib is an FDA-approved, oral irreversible HER1, HER2, and HER4 tyrosine kinase inhibitor that requires loperamide prophylaxis due to the high incidence of severe diarrhea

associated with the drug [6]. However, prophylaxis and management guidelines using antimotility drugs, such as loperamide, an opiate agonist, are empirical interventions to reduce the incidence and/or severity of diarrhea. Strategies for the management of CTD are needed with physiologically relevant antidiarrheal agents, which would improve stool consistency, rather than reduce gastrointestinal motility, preferably as prophylaxis, as the chronic and/or prophylactic use of antimotility drugs has been inadequate or ineffective and not approved by any regulatory agencies.

For instance, the ExteNET study evaluated neratinib-induced diarrhea in 2840 patients without loperamide prophylaxis and showed that the overall diarrhea incidence was 96%, and grade 3/4 diarrhea occurred in 40% of patients with breast cancer. For sake of clarity, Grade 3 diarrhea defined by the CTCAE is passage of >7 loose/watery stools and Grade 4 diarrhea results in hospitalization due to dehydration. Diarrhea-induced permanent neratinib discontinuation was reported in 17% of patients [19, 20].

Subsequently, another study to mitigate neratinib-induced diarrhea called CONTROL study, [21] was designed to prospectively evaluate multiple strategies for managing neratinib-induced diarrhea, including gradual neratinib dose escalation, concomitantly with oral doses of loperamide, budesonide, and/or colestipol, in different arms of HER2-positive breast cancer patients. In the loperamide prophylaxis arm, the rate of grade 3 diarrhea remained at 30%, and the diarrhea-induced discontinuation rate was reported in 20% of patients in this study. Thus, management of neratinib-induced diarrhea with loperamide was at best similar to no prophylaxis (from the ExteNET study) and caused more severe constipation, with an all-grade rate of constipation incidence of 57%. A schedule of concomitant loperamide for 56 days is now required by the FDA label for neratinib treatment.

However given the results from the CONTROL study, the lack of adequate management of diarrhea highlights the need for better antidiarrheal drugs (rather than antimotility drugs) to allow prophylactic and/or treatment regimens for the management of neratinib-induced diarrhea The consequences of CTD burden from neratinib treatment also result in neratinib dose reduction or early termination of neratinib treatment. Increased hospitalization and medical costs for fluid and electrolyte replacement may be required for patients with severe diarrhea [2].

The objective of this study was to evaluate the prophylactic effects of crofelemer in female beagle dogs experiencing neratinib-induced diarrhea without any concomitant loperamide administration, following oral daily dosing of neratinib and either placebo capsules or crofelemer delayed-release tablets (twice or four times daily) for 28 consecutive days. Female dogs were chosen for this study because neratinib is FDA-approved and almost exclusively administered to women with ER, PR, and HER2+ early-stage breast cancer.

## Materials and methods

### Animals

Twenty-four healthy female beagle dogs (*Canis familiaris*) between 8 and 9 months of age, weighing 6.4–8.5 kg, were acquired (Marshall Farms, North Rose, NY, USA). All study participants underwent quarantine and acclimation at a Good Laboratory Practice (GLP) Testing Facility called Comparative Biosciences, Inc. (CBI; Sunnyvale, CA) for 21 days. During the acclimation period, the dogs were observed at least once daily for any abnormal clinical signs. All dogs had normal behavior and had unique ear tattoo numbers. For the pilot study, three dogs were used. For the definitive study, twenty-four dogs were used.

## Study design

A pilot study was conducted to establish a tolerable neratinib dose, that resulted in passage of loose/watery stools (i.e. diarrhea) without causing life threatening side effects. Diarrhea is defined as the passage of loose/watery stools and the clinical definition does not include the mass or volume of loose/watery stool output. Stool volume or mass cannot be measured because under the cage of each dog it is always a combination of urine and stool output. Three female dogs were administered 40 mg to 80 mg of neratinib orally, once daily in the morning, for 14 days. Neratinib doses were adjusted daily for each dog individually to consistently produce diarrhea but to avoid severe side effects such as lack of appetite or having profuse diarrhea causing severe dehydration that may need constant intravenous fluids. Twice daily clinical observations included mortality, morbidity, and overt signs of toxic or pharmacologic effects. Based on this pilot study, dose escalation of neratinib from 40 mg on Day 1 to 80mg on Day 5 was effective in inducing diarrhea without significantly deleterious side effects in female beagle dogs. The pilot study determined that female beagle dogs could tolerate up to 80mg of neratinib.

The definitive 28-day study evaluating the effects of prophylaxis with crofelemer tablets or placebo capsules on neratinib-induced diarrhea in beagle dogs was based on the pilot study; wherein dogs received one neratinib tablet (40mg) for the first 5 days, followed by two neratinib tablets (80 mg) daily with concomitant administration of either placebo capsules four times a day (Control = CTR) or crofelemer 125 mg delayed-release tablets twice (BID) or four times (QID) for the remainder of the 28-day (4 weeks) treatment period.

A total of twenty-four (24) healthy female beagle dogs were used for the definitive study. Dogs were randomized into three treatment groups of eight dogs each: one group of eight dogs received oral neratinib 40–80 mg together with placebo capsules four times per day (CTR), a second group of eight dogs received oral neratinib 40–80 mg together with one crofelemer 125 mg delayed-release tablet twice daily (BID group), and a third group of eight dogs received oral neratinib 40–80 mg together with one crofelemer 125 mg delayed-release tablets four times per day (QID group). Neratinib tablets are available as 40 mg tablets (Nerlynx®) and were provided by Puma Biotechnology, Inc. Crofelemer tablets are available as 125 mg delayed-release tablets (Mytesi®). Crofelemer tablets and placebo capsules were provided by Napo Pharmaceuticals Inc. The experimental unit was a single animal. Facility size and justifiable animal usage per IACUC guidelines limited the study size to a total of 24 female dogs.

## Housing and husbandry

Temperature and relative humidity of the animal rooms were maintained at 61-81˚F and 30–70%, respectively. Animal Welfare Act (USDA, 2013) regulations require that indoor housing at laboratory animal research facilities must have adequate controls such that temperatures remain within the range specified for the species being housed. Humidity was maintained at levels of 30 to 70 percent and ventilation must be sufficient to "minimize odors, drafts, ammonia levels, and moisture condensation". Ambient temperature, or the air temperature surrounding the animal, must not fall below 10˚C (50˚F) for dogs not acclimated to lower temperatures or rise above 29.5˚C (85˚F) for dogs in indoor facilities. These environmental parameters were monitored and recorded daily. Twelve hours of light and twelve hours of dark were provided in the animal rooms. A fluorescent light source was used, with lights turned on at approximately 0700 hours and turned off at approximately 1900 hours each day. LabDiet® 5006 Laboratory Canine Diet (Purina Mills, Inc., St. Louis, MO) was provided according to package labeling throughout the acclimation and in-life phases. Lot number(s) and Certificate(s) of Analysis (as applicable) were maintained by CBI. Fresh water from the

Municipal Water Supply of Sunnyvale, California was provided *ad libitum* to the animals via a rack watering system and water bowls. The water supply was periodically monitored for chlorine content and bacterial contamination. The results of these analyses are maintained on file at CBI. Animals were housed individually in stainless steel cages with raised floors in a room dedicated to dogs. General procedures for animal housing and husbandry were conducted according to CBI SOPs and met all regulations concerning use of animals in research, including the U.S. Department of Agriculture regulations (9 CFR Chapter 1) implementing the Animal Welfare Act (7 USC 2131 *et seq.*) and the recommendations of the National Research Council's "Guide for Care and Use of Laboratory Animals" (National Academy Press, 2011).

## Dose administration, animal care, and monitoring

There were no inclusion or exclusion criteria set for enrollment of healthy female beagle dogs into this study. Neratinib dosing was planned as defined by the pilot study.

All animals in this study were clinically evaluated once daily in the morning by an accredited and licensed veterinarian blinded to the study treatments. Clinical observations, including morbidity, mortality, and overt signs of toxic or pharmacologic effects were recorded for all dogs four times daily throughout the 28-day study.

Neratinib dose reductions or 1-day holidays were allowed when an individual study animal was deemed to be dehydrated, lethargic, or inappetent. Dose reductions were classified as dogs receiving 40mg of neratinib (1 tablet) instead of 80mg (2 tablets), while dose holidays allowed for 1-day only when animals were deemed incapable of receiving either 80mg or 40mg of neratinib. Data regarding neratinib tablets administration was collected and analyzed to avoid unbalanced dosages of neratinib between treatment groups.

## Data collection

All animals in this study were clinically evaluated four times daily. Body weight (kg) was collected prior to dosing, daily thereafter, and prior to necropsy. Food consumption was measured daily qualitatively. Food consumption was assigned a 3-point scale. Dogs not consuming any food during the day were assigned a point equivalent to zero, dogs consuming some of its food (equal or less than 50% of the food provided) were assigned one point, and dogs consuming most of the food provided (more than 50% of the food provided) were given two points. Water was available *ad libitum* but water consumption was not measured.

Blood samples were collected on Day 0, Day 7, Day 14, Day 21, and Day 28 for clinical pathology analysis (clinical chemistry and hematology). At termination a full necropsy was conducted, organ weights were recorded, and a complete set of tissues was collected and fixed for histopathologic evaluation. The gastrointestinal tract was removed and evaluated for gross lesions. Tissue samples for histopathology were collected from the stomach, duodenum, jejunum, ileum, cecum, colon, and rectum.

Hydration status was assessed daily. Dehydration due to neratinib-induced diarrhea was an expected side effect. In order to minimize cases of severe dehydration, subcutaneous fluid administration was chosen as a "rescue" protocol to treat mild and moderate cases of dehydration. Dehydration was classified as mild (minimal loss of skin turgor, semi-dry mucous membranes, normal eyes), moderate (moderate loss of skin turgor, dry mucous membranes, weak rapid pulses, enophthalmos), and Severe (considerable loss of skin turgor, severe enophthalmos, tachycardia, extremely dry mucous membranes, weak and thready pulses, hypotension, lethargy). Cases of mild and moderate dehydration were treated with 150mL or 300mL of Lactated Ringer's solution administered subcutaneously, respectively. Severe dehydration would

require intravenous fluids. However, severe dehydration was not observed in any dogs during the 4-week study period.

Stool consistency was evaluated twice-daily during the 28-day definitive study period and scored using the well-established 7-point scale known as the Purina Fecal Scoring (PFS; **S1 Fig**), and this was the only fecal scoring system used throughout the study.

## Endpoints

The objective of this 28-day study was to assess the safety and effectiveness of prophylaxis with daily oral doses of crofelemer tablets or placebo capsules, concomitantly with daily neratinib dosing, in reducing neratinib-induced diarrhea without the use of any dose of the antimotility drug, loperamide. Stool consistency was recorded using the 7-point scale Purina Fecal Scoring (PFS) system and the scoring of stools was further dichotomized into loose and/or watery stools by combining stools scored as a 6 and 7, which were defined as loose/watery stools, while all stools scored between 1 and 5 were considered non-diarrheic. The average weekly number of loose/watery stools were analyzed across the three treatment groups.

Additionally, to represent the number of dogs that responded to crofelemer treatment in resolving neratinib-induced diarrhea, the definition of a responder dog was created. A responder was defined *a priori*, as any dog with an average of $\leq 1$ loose/watery stool per day (or $\leq 7$ loose/watery stools per week) for at least 2 of the 4 weeks of the 4-week study, as this is deemed clinically meaningful and is considered Grade 1 (mild) diarrhea.

Furthermore, because neratinib is used in humans with loperamide prophylaxis from the onset of treatment for at least 56 days [6], the hypothesis in this study was that dogs in the groups receiving concomitant crofelemer tablets (BID or QID) with neratinib would provide greater prophylaxis from neratinib-induced diarrhea and a greater proportion of dogs would be responders over the 4-week study period.

This responder analysis evaluated the weekly complements Purina of fecal scores (6 or 7) to define a percentage of dogs in each group that achieved <7 loose/watery stools each week and were deemed to be "clinical successes" for the prophylaxis of neratinib-induced diarrhea.

Finally, because dose reduction and dose holidays for neratinib were allowed during any of the 28 days of this study, the number of weekly loose/watery stools per daily dose of the number of neratinib tablets was also evaluated as one of the endpoints in this study.

Average daily number of neratinib tablets administered, weekly body weight changes (from baseline), qualitative assessment of food intake, and volume of fluids required to maintain euhydration were also evaluated.

## Ethical statement

This study was carried out in strict accordance with the recommendations in the Guide for the Care and Use of Laboratory Animals of the National Institutes of Health. The protocol was approved by the Institutional Animal Care and Use Committee (IACUC) of Comparative Biosciences, Inc. (IACUC Proposal Number CB19-8011-D-EF) on 26 February 2020 prior to the start of the study.

Animals were euthanized on Day 28. Animals were first sedated with ketamine/diazepam (5/0.2 mg/kg, IV) and blood was drawn for clinical pathology. Euthanasia was then performed using a commercial euthanasia solution (Euthasol®, pentobarbital sodium and phenytoin sodium, Delmarva Labs) diluted with an equal volume of sterile saline. The solution was administered intravenously, approximately 150 mg/kg or to effect.

## Statistical methods

Data from all animals were included in the statistical analysis and no data were excluded. For all analyses, data from all 8 animals for each of the three experimental treatment groups (CTR i.e. neratinib + placebo capsules; or neratinib + crofelemer tablets BID or QID) were analyzed. The analysis was performed on a weekly basis for all study weeks; this included weekly average over the 4-week study period; as well as weekly average number of loose/watery stools for each of the individual study weeks (week 1, week 2, week 3, and week 4) separately.

Analysis of Covariance (ANCOVA) was used to analyze the prophylactic effect of crofelemer in reducing the average number of weekly loose/watery stools, as well as the average number of daily neratinib tablets per week, weekly body weight changes (from baseline), qualitative food intake, and volume of fluids administered over the 4-week study period.

Least Square Means (LSM) of the average weekly number of loose/watery stools were analyzed ± SD (standard deviation). Least square means were adjusted for baseline fecal scores and body weight as covariates. Normality of residuals was assessed using the Shapiro-Wilk statistic for closeness to normality. The Shapiro-Wilk statistics for the analyses were all > 0.9, indicating that no data transformation was needed, and the untransformed summary statistics were adequate. All procedures were conducted using SAS 9.3 (Cary, NC, USA).

The effect of crofelemer in reducing the average number of weekly loose/watery stools while controlling for the number of neratinib tablets administered were also analyzed using ANCOVA. The number of neratinib tablets and the baseline fecal scores were used as covariates.

For the responder analysis, the differences in the percentage of dogs that were responders for either 2 out of 4 weeks, or each of the four weeks, to the diarrhea prophylaxis was conducted using an unconditional exact binomial test to evaluate the proportion of responder dogs in each group.

The power of the study was computed using standardized effect sizes (mean difference from CTR divided by pooled SD). For clarity, effect sizes can be judged to be large if they are close to or greater than 1.0, as the sample sizes required for adequate power in future studies are generally small. For example, for 90% power (alpha = 0.05 2-sided), N = 34, 27, and 23 per treatment group would be required if the TRUE underlying effect sizes were 0.8, 0.9, and 1.0, respectively.

## Results

Daily dosing of neratinib caused diarrhea in beagle dogs. Purina fecal scores were collected daily for all 28 days (4 weeks) for all dogs for each bowel movement. Stool consistency was dichotomized into Loose/Watery stools (defined as scores 6 and/or 7 by the Purina Fecal Score) and non-diarrheic (scores 1 through 5 on the Purina Fecal Score). Dogs were dosed 40 mg or neratinib orally once daily for the first 5 days followed by daily doses of 80 mg for 23 days concomitantly with placebo capsules QID or crofelemer tablets BID or QID.

Prophylaxis of neratinib-induced diarrhea with crofelemer tablets significantly reduced the average number of weekly loose/watery stools (**Table 1**) when compared to dogs receiving neratinib with placebo capsules. Over the entire 4-week study period, dogs in the CTR group, that received neratinib with placebo capsules, had a significantly higher number of average number of weekly loose/watery stools (i.e. worse diarrhea) when compared to those dogs receiving neratinib with either crofelemer BID (8.70 ± 2.2 vs. 5.96 ± 2.2, p = 0.028); or neratinib with crofelemer QID (8.70 ± 2.2 vs. 5.74 ± 2.2, p = 0.022). There was no difference between the crofelemer BID or QID groups on reducing neratinib-induced diarrhea (p = 0.84), suggesting

**Table 1. Average number of loose/watery stools per week by treatment group over the 4-week crofelemer study period in neratinib-induced diarrhea in dogs (n = 8 per treatment group).** Diarrhea was induced by daily doses of neratinib, and fecal scores using the Purina Fecal Scoring (PFS) System were collected daily for 28 days (4 weeks). Dogs were dosed with 40 mg of neratinib orally once daily for the first 5 days followed by daily doses of 80 mg of neratinib for 23 days. Stool consistency was dichotomized into loose/watery stools (defined as scores of 6 or 7 on the PFS) and non-diarrheic (scores 1 through 5 on the PFS).

| Average number of loose/watery stools per week | Treatment Groups | Least Square Means (LSM) | Standard Deviation (SD) | P-values | |
|---|---|---|---|---|---|
| | | | | Active vs Control | BID vs QID |
| **Combined 4 weeks** | Control | 8.71 | 2.18 | - | - |
| | Crofelemer BID | 5.96 | 2.18 | 0.03* | - |
| | Crofelemer QID | 5.74 | 2.18 | 0.02* | 0.84 |
| **Week 1** | Control | 7.54 | 2.77 | - | - |
| | Crofelemer BID | 4.36 | 2.77 | 0.04* | - |
| | Crofelemer QID | 4.35 | 2.77 | 0.05* | 0.99 |
| **Week 2** | Control | 11.15 | 3.19 | - | - |
| | Crofelemer BID | 9.37 | 3.19 | 0.30 | - |
| | Crofelemer QID | 8.23 | 3.19 | 0.11 | 0.49 |
| **Week 3** | Control | 8.69 | 3.74 | - | - |
| | Crofelemer BID | 6.61 | 3.74 | 0.31 | - |
| | Crofelemer QID | 6.21 | 3.74 | 0.24 | 0.83 |
| **Week 4** | Control | 7.45 | 2.84 | - | - |
| | Crofelemer BID | 3.52 | 2.84 | 0.02* | - |
| | Crofelemer QID | 4.16 | 2.84 | 0.05* | 0.66 |

Treatment groups were defined as a placebo-controlled group (CTR) receiving placebo capsules orally four times a day, crofelemer (125mg) administered orally twice daily (BID), and crofelemer (125mg) administered orally four times a day (QID) for 28 days.

* $p \leq 0.05$

that prophylaxis of neratinib-induced diarrhea could be achieved with crofelemer BID oral dosing.

Over all 4 weeks of the study, the proportion of responder dogs that were able to maintain their stool frequency at $\leq 1$ loose/watery stools per day (i.e. $\leq 7$ per week) were found to be 63% (5/8 dogs) in the crofelemer BID group (p<0.05), and 75% (6 out of 8 dogs) in the crofelemer QID group (p<0.02), whereas only 12.5% (1/8 dogs) in the CTR group achieved the responder definition. There was no difference between the crofelemer BID and QID treatment groups. (**Fig 1A**). Additionally, animals that were considered to be a responder for at least half of the study period (i.e. for 2 out of 4 weeks) were found to be 100% (8/8 dogs) in the crofelemer BID group (p<0.03) and 87% (7/8 dogs) in the crofelemer QID group (p = 0.14), whereas only 50% (4/8 dogs) in the CTR group achieved the responder definition. There was no difference between the crofelemer BID and QID treatment groups. These results are presented in **Fig 1B**. This 28-day study also showed that female beagle dogs were able to tolerate a consistent daily dose of neratinib during the 28-days, with no differences in neratinib tablet administration between any of the treatment groups at any point during the study (**Table 2**). Over the entire 4-week study period, dogs in the CTR group were administered on average 1.23±0.15 tablets of neratinib/day; while dogs in the crofelemer BID and QID were administered 1.32 ±0.15 and 1.30±0.15 neratinib tablets per day, respectively (p>0.05).

When controlling for the daily neratinib dose, the crofelemer BID and QID treated groups had a significant reduction in the average number of weekly loose/watery stools on week 2, while a statistical trend was noted over the entire 4-week study period (**Table 3**). Specifically in week two, dogs receiving neratinib and crofelemer (BID or QID) had significantly reduced the average number of weekly loose/watery stools, compared to those dogs receiving neratinib and

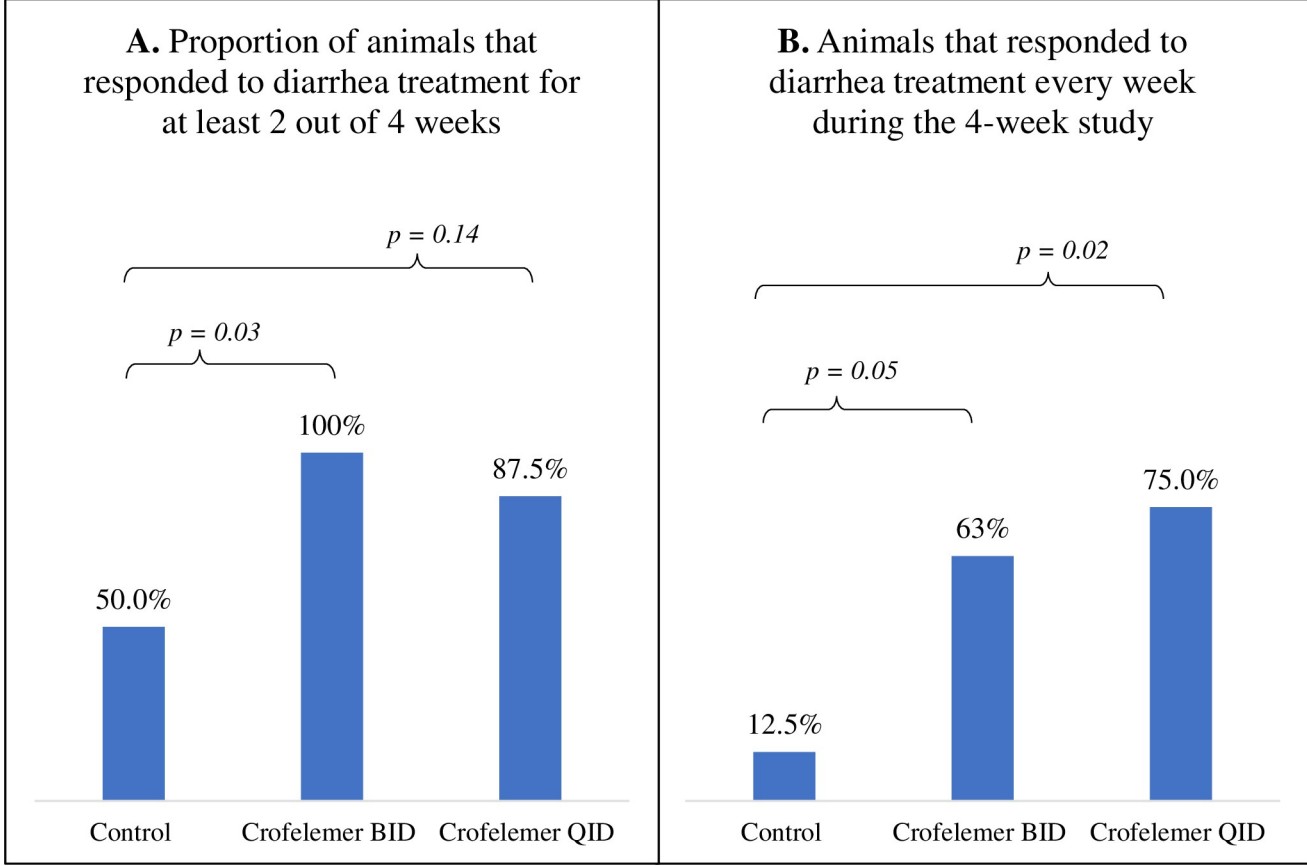

**Fig 1. Proportion of dogs that responded to crofelemer treatment in resolving neratinib-induced diarrhea by treatment group over the 4-week crofelemer study period in neratinib-induced diarrhea in dogs (n = 8 per treatment group).** A responder dog (as defined *a priori*) was any dog with an average of $\leq$ 7 loose/watery stools per week, which equals $\leq$ 1 loose/watery stool per day, for at least 2 weeks of the 4-week study period (**1A.**). Additionally, results are presented for animals that were a responder for every week of the 4-week study period (**1B.**). Treatment groups were defined as a placebo-controlled group (CTR) receiving placebo capsules orally four times a day, crofelemer (125mg) administered orally twice daily (BID), and crofelemer (125mg) administered orally four times a day (QID) for 28 days. * p < 0.05.

placebo capsules in the CTR group. The average number of loose/watery stools in week 2 for the crofelemer BID group compared to CTR group as 4.66 ± 0.7 vs. 3.36 ± 0.7 (p = 0.002). Similarly, comparing CTR group with crofelemer QID group, the average number of loose/watery stools were 4.66 ± 0.7 vs. 3.34 ± 0.7 (p = 0.002). No differences were found between crofelemer BID and QID (p = 0.96). Across the entire 4 week period, dogs receiving neratinib with either crofelemer BID or QID tended to have lower weekly average number of loose/watery stools when compared to dogs receiving neratinib and placebo capsules in the CTR group (4.89 ± 0.7 vs. 5.62 ± 0.7 p = 0.05, and 4.85 ± 0.7 vs. 5.62 ± 0.7 p = 0.06, respectively).

In all groups and animals, the pathological evaluation of gastrointestinal tissues showed no significant changes among the dogs receiving neratinib with placebo capsules or crofelemer. Histopathological evaluation showed that the mucosa, submucosa, muscularis, and serosa were unremarkable, and there was no evidence of villus blunting, inflammation, necrosis, increased apoptosis, crypt damage, thrombosis, or other lesions in any of the dogs. These findings indicate that daily administration of neratinib induced a severe clinical presentation of diarrhea, intestinal hemorrhage, and dehydration; however, these clinical symptoms did not produce mucosal changes. Preliminary results from this neratinib-crofelemer study have been previously reported [22].

**Table 2. Average daily number of neratinib tablets administered each week by treatment group over the 4-week crofelemer study period in neratinib-induced diarrhea in dogs (n = 8 per treatment group).** Diarrhea was induced by daily doses of neratinib. Dogs were dosed with one tablet (40 mg) of neratinib orally once daily for the first 5 days followed by once daily doses of two tablets (80 mg) of neratinib for the remaining 23 days of the 28-day study. Treatment groups were defined as a placebo-controlled group (CTR) receiving placebo capsules orally four times a day, crofelemer (125mg) administered orally twice daily (BID), and crofelemer (125mg) administered orally four times a day (QID) for 28 days.

| Average daily number of neratinib tablets administered each week | Treatment Groups | Least Square Means (LSM) | Standard Deviation (SD) | P-values | |
|---|---|---|---|---|---|
| | | | | Active vs Control | BID vs QID |
| **Combined 4 weeks** | Control | 1.23 | 0.15 | - | - |
| | Crofelemer BID | 1.32 | 0.15 | 0.25 | - |
| | Crofelemer QID | 1.30 | 0.15 | 0.37 | 0.80 |
| **Week 1** | Control | 1.22 | 0.07 | - | - |
| | Crofelemer BID | 1.29 | 0.07 | 0.08** | - |
| | Crofelemer QID | 1.27 | 0.07 | 0.18 | 0.67 |
| **Week 2** | Control | 1.32 | 0.39 | - | - |
| | Crofelemer BID | 1.49 | 0.39 | 0.42 | - |
| | Crofelemer QID | 1.53 | 0.39 | 0.33 | 0.83 |
| **Week 3** | Control | 1.11 | 0.36 | - | - |
| | Crofelemer BID | 1.15 | 0.36 | 0.84 | - |
| | Crofelemer QID | 1.13 | 0.36 | 0.95 | 0.89 |
| **Week 4** | Control | 1.25 | 0.19 | - | - |
| | Crofelemer BID | 1.36 | 0.19 | 0.28 | - |
| | Crofelemer QID | 1.28 | 0.19 | 0.76 | 0.40 |

Treatment groups were defined as a placebo-controlled group (CTR) receiving placebo capsules orally four times a day, crofelemer (125mg) administered orally twice daily (BID), and crofelemer (125mg) administered orally four times a day (QID) for 28 days.

** $p \leq 0.1$

There were no significant changes in the body weights of dogs receiving neratinib with placebo (CTR) or neratinib with crofelemer BID or neratinib with crofelemer QID over the 28-day treatment period. (S1 Table)

Qualitative food consumption is presented in S2 Table. Water was available *ad libitum* to all dogs, although water consumption was not measured in this study.

The weekly average volume of subcutaneous fluid administration each week by treatment group is shown in S3 Table. There were no differences in the volumes of subcutaneous fluids administered among any treatment groups at any time during the study.

Daily clinical exams were performed during the study period. Dogs in all three treatment groups had similar side effect profile throughout the 4-week study. Emesis, hematochezia, sialorrhea, lack of appetite, nausea and lethargy were similar between treatment groups and data is presented in S4 Table. The number of dogs with out-of-range clinical chemistry values by treatment group on Study Days 0, 7, 14, 21, and 28 are presented in S5 Table. Clinical chemistry values were generally unremarkable on Day 0. Mild hypoalbuminemia, hypoproteinemia, and hypocalcemia were noted in most to all animals in all treatment groups on Day 7, 14, 21, and 28 (S5 Table). In general, most abnormal values were only slightly elevated or decreased compared to normal ranges. There were no differences in clinical chemistry values between CTR dogs and crofelemer BID and QID dogs. The number of dogs with out-of-range hematology values by treatment group on Study Days 0, 7, 14, 21, and 28 are also presented in S5 Table.

The daily raw body weights for all dogs over the 28-day treatment period are presented in S6 Table. The daily qualitative food consumption scores for all dogs over the 28-day treatment period are presented in S7 Table. The daily Purina Fecal Scores (PFS) and number of bowel

**Table 3. Average number of loose/watery stools per week per neratinib dose by treatment group over the 4-week crofelemer study period in neratinib-induced diarrhea in dogs (n = 8 per treatment group).** Diarrhea was induced by daily doses of neratinib, and fecal scores using the Purina Fecal Scoring (PFS) System were collected daily for 28 days (4 weeks). Dogs were dosed 40 mg or neratinib orally once daily for the first 5 days followed by daily doses of 80 mg for 23 days. Stool consistency was dichotomized into loose/watery stools (defined as scores of 6 or 7 on the PFS) and non-diarrheic (scores of 1 through 5 on the PFS).

| Average number of loose/watery stool per neratinib dose per week | Treatment Groups | Least Square Means (LSM) | Standard Deviation (SD) | P-values Active vs Control | P-values BID vs QID |
|---|---|---|---|---|---|
| Combined 4 weeks | Control | 5.62 | 0.69 | - | - |
| | Crofelemer BID | 4.89 | 0.69 | 0.06** | - |
| | Crofelemer QID | 4.85 | 0.69 | 0.05* | 0.91 |
| Week 1 | Control | 4.66 | 0.68 | - | - |
| | Crofelemer BID | 3.36 | 0.68 | 0.002* | - |
| | Crofelemer QID | 3.34 | 0.68 | 0.002* | 0.96 |
| Week 2 | Control | 5.73 | 2.35 | - | - |
| | Crofelemer BID | 4.76 | 2.35 | 0.45 | - |
| | Crofelemer QID | 4.36 | 2.35 | 0.30 | 0.73 |
| Week 3 | Control | 5.41 | 4.09 | - | - |
| | Crofelemer BID | 4.89 | 4.09 | 0.81 | - |
| | Crofelemer QID | 6.30 | 4.09 | 0.69 | 0.50 |
| Week 4 | Control | 4.32 | 1.37 | - | - |
| | Crofelemer BID | 2.76 | 1.37 | 0.04* | - |
| | Crofelemer QID | 3.24 | 1.37 | 0.16 | 0.49 |

Treatment groups were defined as a placebo-controlled group (CTR) receiving placebo capsules orally four times a day, crofelemer (125mg) administered orally twice daily (BID), and crofelemer (125mg) administered orally four times a day (QID) for 28 days.

* p ≤ 0.05

** p ≤ 0.1

movements for all dogs over the 28-day treatment period are presented in S8 Table. Finally, the daily qualitative hydration scores for all dogs over the 28-day treatment period are presented in S9 Table.

## Discussion

Crofelemer is a novel antisecretory antidiarrheal drug that reduces intestinal chloride ion and fluid secretion through partial antagonism and use-dependent modulation of CFTR and CaCC channels in the apical membrane of the intestinal mucosa, to normalize the fluid and electrolyte balance in the gastrointestinal (GI) tract. The antisecretory mechanism of crofelemer has been studied in cell lines to evaluate its effects on chloride ion secretion in CFTR and CaCCs [7]. Crofelemer produces an extracellular voltage-independent block of the CFTR channel while stabilizing CFTR in the closed state, and its pharmacodynamic effects on CFTR are prolonged after exposure. Crofelemer also produces a voltage-independent block of CaCC, which is structurally unrelated to the CFTR. Neratinib is an oral irreversible inhibitor of EGFR/HER1, HER2, and HER4 tyrosine kinases that causes severe diarrhea and presents management issues for antidiarrheal treatment and/or prophylaxis to ensure the continuity of breast cancer treatment. Neratinib, an irreversible pan-HER tyrosine kinase inhibitor, causes secretory diarrhea resulting from excess chloride ions and fluid secretion into the intestinal lumen through hyperactivation of the apical CFTR chloride ion channel [19, 20]. This chloride ion secretion into the gut lumen may be cooperative with the activation of the calcium-activated chloride channel anoctamin1 (ANO1). Targeting excessive chloride ion secretion via

administration of crofelemer represents a novel 'targeted' antidiarrheal intervention for the management of neratinib-induced diarrhea.

Induction of diarrhea by oral neratinib in female beagle dogs and concomitant administration of crofelemer or placebo tablets were well tolerated over the 28-day study period, as there were no differences in the deleterious effects of the model on body weight changes, hydration status, or other hematological or clinical chemistry parameters among the three treatment groups. The initial average body weight difference between dogs enrolled in the crofelemer BID and QID groups were a product of randomization without blocking for body weight, and most importantly, no differences between the CTR group and crofelemer QID or BID treatment groups were found. Furthermore, this initial body weight difference did not affect neratinib dosage and fluid administration between crofelemer BID and QID, but most importantly no body weight changes were found throughout the study between the crofelemer BID and QID treatment groups.

Furthermore, the average weekly number of loose/watery stools was significantly improved in dogs receiving neratinib with either crofelemer BID or QID compared to the dogs receiving neratinib with placebo capsules (Table 1). Additionally, the number of neratinib doses administered per week across the three treatment groups was not significantly different across all three treatment groups (Table 2), suggesting that the differences in the weekly number of loose/watery stools per week between the CTR group and the crofelemer BID and QID groups was not due to reduction in the daily dose of neratinib administered. Furthermore, as shown in S3 Table, the volume of subcutaneous fluids administered per week by treatment group was not different between any of the three treatment groups at any time during the study. Additionally, there were no significant differences in body weights between treatment groups during the study (S1 Table), the amount of food consumption between treatment groups did not differ during the study (S2 Table), and there were no significant differences in clinical chemistry or hematology parameters between treatment groups during the study (S5 Table).

Thus, prophylaxis for neratinib-induced diarrhea with crofelemer tablets administered orally either BID or QID, reduced the average number of weekly loose/watery stools when compared to dogs receiving neratinib with placebo capsules. The hydration and clinical status of all dogs was maintained throughout the study allowing equivalent daily dosing of neratinib and there were no significant differences in the amount of subcutaneous fluids administered, or body weights, food consumption, and clinical pathology among the three treatment groups during the study.

Potential pathophysiological mechanisms for CTD, including those from neratinib, are complex and involve several overlapping inflammatory, secretory, and neural mechanisms, as well as acute mucosal damage with altered tissue architecture and barrier function. A proposed mechanism of targeted therapy drugs that induce gastrointestinal toxicity is based on EGFR expression in normal gastrointestinal mucosa and its role in the regulation of intestinal chloride ion secretion [23, 24]. A recent study in rats with osimertinib- and afatinib-induced diarrhea showed an increase in stool water content and that diarrhea was attenuated by intraperitoneal treatment with an experimental calcium-activated chloride ion channel (CaCC) inhibitor (CaCCinh-A01) [25].

Crofelemer elicits its effects on apical membrane transport and signaling processes involved in intestinal chloride ion secretion by a novel use-dependent inhibitory modulation mechanism. Crofelemer reduces chloride ion secretion by the CFTR channel through use-dependent inhibitory modulation of CFTR, resulting in an IC50 of 6–7 μM [7]. Crofelemer's action resisted washout, with inhibition lasting several hours after washout. Crofelemer was also found to inhibit the intestinal calcium-activated chloride channel ANO1, also known as TMEM16A and DOG, via a voltage-independent inhibition mechanism with a maximum

inhibition of 90% and an IC50 of 6.5 μM. The dual inhibitory modulation by crofelemer on two structurally unrelated intestinal chloride ion channels results in its unique physiological intestinal antisecretory antidiarrheal effects.

In the present study, female beagle dogs received neratinib orally daily concomitantly with either placebo capsules (CTR) or crofelemer 125 mg delayed-release tablet two or four times/day (BID or QID) for 28 consecutive days. Only 12.5% of dogs in the control group were able to maintain their loose/watery stools to equal or less than 7 per week, while 63% and 75% dogs receiving crofelemer BID or QID, respectively, were able to respond over this 4-week study. The average number of weekly loose/watery stools were significantly lower in the dogs that received concomitant neratinib with either crofelemer BID or QID; compared to those dogs receiving neratinib with placebo capsules. The average number of weeks with no loose/watery stools was also higher for dogs receiving neratinib with crofelemer BID or QID concomitantly. Weekly mean stool consistency scores were improved with more formed stools in dogs receiving neratinib with crofelemer BID or QID compared to those receiving neratinib with placebo capsules. Most importantly, no dogs needed to receive loperamide, an antimotility drug, either prophylactically or as a "rescue medication" over the 4-week study period, as dogs in all groups were allowed to be rescued with subcutaneous fluids and/or neratinib dose reduction or dose interruption.

This is a first-of-its-kind study that evaluated the prophylactic effects of daily crofelemer dosing in female beagle dogs with moderate-to-severe diarrhea induced by daily oral neratinib administration without any concomitant use of loperamide. A limitation of this study is the 28-day duration of oral administration of neratinib and crofelemer or placebo. Neratinib is intended to be administered daily to HER-2 positive breast cancer patients on a chronic basis, i.e.–every day for as long as possible. The 28-day study period in our study represents a more limited severity or incidence of diarrhea than that experienced by human patients receiving neratinib. Thus, the effect size estimated in our study may represent the lower end of expected improvement in the effect size for breast cancer patients receiving neratinib with crofelemer in a future human clinical trial. Given the similarities in the gastrointestinal mucosa between humans and dogs, the results of this study provide a preclinical proof-of-concept that supports the ongoing phase 3 prophylactic crofelemer clinical study in adult human patients with solid tumors receiving targeted therapy with or without standard chemotherapy (NCT04538625; OnTarget study). The results from our prophylactic crofelemer- neratinib dog study also buttress the results observed in a prophylactic crofelemer study in HER2-positive breast cancer patients receiving trastuzumab, pertuzumab, with a taxane, with or without carboplatin, called the HALT-D study [9].

The HALT-D study suggested that patients receiving prophylaxis with crofelemer had a reduction in the severity of higher-grade diarrhea, and patients receiving crofelemer prophylaxis also had higher odds of having diarrhea resolution compared to those receiving HER2-targeted therapy with standard chemotherapy using the current standard of care (SOC) treatments to manage their diarrhea. Crofelemer is currently being investigated for the prophylaxis of diarrhea in a randomized, double-blind, placebo-controlled phase 3 clinical trial for adult patients with solid tumors receiving targeted cancer therapies with or without standard chemotherapy regimens, evaluating the average weekly number of loose and/or watery stools over the entire 12-week study period as a continuous endpoint for the safety and efficacy of the drug (NCT04538625; OnTarget study).

## Supporting information

**S1 Fig. Scoring chart for the Purina Fecal Score (PFS) system.**
(TIF)

**S1 Table. Body weight at enrollment and body weight changes per week by treatment group over the 4-week crofelemer study period in neratinib-induced diarrhea in dogs.**
(DOCX)

**S2 Table. Food consumption per week by treatment group over the 4-week crofelemer study period in neratinib-induced diarrhea in dogs.**
(DOCX)

**S3 Table. Weekly average volume (mL) of subcutaneous fluid administration each week by treatment group over the 4-week crofelemer study period in neratinib-induced diarrhea in dogs.**
(DOCX)

**S4 Table. Adverse event profile from Days 0 through 28 of 4-week crofelemer study period in neratinib-induced diarrhea in dogs.**
(DOCX)

**S5 Table. Number of dogs with out-of-range clinical chemistry and hematology parameters on study Days 0, 7, 14, 21, and 28 by treatment group over the 4-week crofelemer study period in neratinib-induced diarrhea in dogs.**
(DOCX)

**S6 Table. Daily raw body weights by treatment group over the 4-week crofelemer study period in neratinib-induced diarrhea in dogs (n = 8 per treatment group).**
(PDF)

**S7 Table. Daily food consumption by treatment group over the 4-week crofelemer study period in neratinib-induced diarrhea in dogs (n = 8 per treatment group).**
(PDF)

**S8 Table. Daily Purina Fecal Scores and number of bowel movements by treatment group over the 4-week crofelemer study period in neratinib-induced diarrhea in dogs (n = 8 per treatment group).**
(PDF)

**S9 Table. Daily hydration scores by treatment group over the 4-week crofelemer study period in neratinib-induced diarrhea in dogs (n = 8 per treatment group).**
(PDF)

## Acknowledgments

The authors would like to acknowledge the medical expertise of the indigenous communities of the Amazon basin, who discovered the medicinal use of the tree from which the crofelemer was extracted and purified.

## Author Contributions

**Conceptualization:** Michael Guy, Andre Teixeira, James Bolognese, Pravin Chaturvedi.

**Data curation:** Michael Guy, Andre Teixeira, Carol Meschter, Pravin Chaturvedi.

**Formal analysis:** Andre Teixeira, James Bolognese, Pravin Chaturvedi.

**Funding acquisition:** Michael Guy.

**Investigation:** Michael Guy, Carol Meschter.

**Methodology:** Michael Guy, Andre Teixeira, Carol Meschter, James Bolognese.

**Software:** James Bolognese.

**Supervision:** Michael Guy, Andre Teixeira, Pravin Chaturvedi.

**Writing – original draft:** Michael Guy, Andre Teixeira, Allison Shrier, James Bolognese, Pravin Chaturvedi.

**Writing – review & editing:** Michael Guy, Andre Teixeira, Allison Shrier, Carol Meschter, James Bolognese, Pravin Chaturvedi.

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
