## [Decision Letter · Decision Letter 0]

19 Apr 2023

PONE-D-23-05162Effects of orally administered crofelemer on the incidence and severity of neratinib-induced diarrhea in female dogsPLOS ONE

Dear Dr. Guy,

Thank you for submitting your manuscript to PLOS ONE. After careful consideration, we feel that it has merit but does not fully meet PLOS ONE’s publication criteria as it currently stands. Therefore, we invite you to submit a revised version of the manuscript that addresses the points raised during the review process.

It was felt that the authors need to adequately highlight the novelty of these findings and also critically address the issues related to rigorous statistical analysis, inclusion of the requested data (as suggested by the reviewers) and need to add appropriate citations where needed.  In addition, the authors need to address the issue of the **potential**
**conflict of interest** as raised by the Reviewer 2 by clarifying the funding and appointment issues of the authors.

We look forward to receiving your revised manuscript.

Kind regards,

Pradeep Dudeja

Academic Editor

PLOS ONE

Journal Requirements:

Reviewers' comments:

Reviewer's Responses to Questions

**Comments to the Author**

1. Is the manuscript technically sound, and do the data support the conclusions?

Reviewer #1: Partly

Reviewer #2: Partly

2. Has the statistical analysis been performed appropriately and rigorously? 

Reviewer #1: Yes

Reviewer #2: No

3. Have the authors made all data underlying the findings in their manuscript fully available?

Reviewer #1: Yes

Reviewer #2: No

4. Is the manuscript presented in an intelligible fashion and written in standard English?

Reviewer #1: Yes

Reviewer #2: Yes

5. Review Comments to the Author

Reviewer #1: In this study, Guy et al. have evaluated orally-administered crofelemer to prevent neratinib-induced diarrhea without using loperamide in a canine model. Neratinib, a pan-HER tyrosine kinase inhibitor that is used to treat breast cancer, causes severe diarrhea in >95% of patients. Crofelemer, a novel intestinal chloride ion channel modulator, is an approved antidiarrheal for patients with HIV receiving antiretroviral therapy. This is an important study as cancer therapy-related diarrhea (CTD) is a clinically-relevant problem. However, the authors need to emphasize the novelty of this concept and should state those clearly.

This is potentially a very important study. Please address these points to improve the manuscript:

1. In this study, the investigators evaluated crofelemer prophylaxis in reducing the incidence and severity of neratinib-induced diarrhea without loperamide in dogs. The drug has been used extensively in HIV patients (below). The authors are requested to identify and stress the differences in regimen, mechanisms, dosage, or surprising reasons for these findings.

2. There is an FDA report of conditional approval of oral tablets to treat chemotherapy-induced diarrhea in dogs to Jaguar Animal Health. Canalevia-CA1 (active ingredient in Canalevia-CA1 is crofelemer) was allowed on December 21, 2021. https://www.fda.gov/news-events/press-announcements/fda-conditionally-approves-first-oral-tablet-treat-chemotherapy-induced-diarrhea-dogs. Please identify and emphasize the differences between these two reports to indicate novelty for the average reader.

3. The authors have rightly identified and stressed, a number of reports in travelers’ diarrhea, HIV-associated diarrhea, diarrhea-prominent IBS, and its mechanisms have been previously reported. However, the number of such studies is large, and there is a possibility of the readers ending up considering this important paper as a “me-too” article. The authors should emphasize the differences on why this clinical improvement in this cohort was unexpected. Many possibilities have already been included in the discussion sections of the following articles:

• Crutchley RD, Miller J, Garey KW. Crofelemer, a novel agent for treatment of secretory diarrhea. Ann Pharmacother 2010;44(5):878-84. doi: 10.1345/aph.1M658. Epub 2010 Apr 13. PMID: 20388859

• Yeo QM, Crutchley R, Cottreau J, Tucker A, Garey KW. Crofelemer, a novel antisecretory agent approved for the treatment of HIV-associated diarrhea. Drugs Today (Barc). 2013 Apr;49(4):239-52. doi: 10.1358/dot.2013.49.4.1947253.PMID: 23616951

• Cottreau J, Tucker A, Crutchley R, Garey KW. Crofelemer for the treatment of secretory diarrhea. Expert Rev Gastroenterol Hepatol. 2012 Feb;6(1):17-23. doi: 10.1586/egh.11.87.PMID: 22149578

• Macarthur RD, Hawkins TN, Brown SJ, Lamarca A, Clay PG, Barrett AC, Bortey E, Paterson C, Golden PL, Forbes WP. Efficacy and safety of crofelemer for noninfectious diarrhea in HIV-seropositive individuals (ADVENT trial): a randomized, double-blind, placebo-controlled, two-stage study.

HIV Clin Trials. 2013 Nov-Dec;14(6):261-73. doi: 10.1310/hct1406-261.

• Mangel AW, Chaturvedi P. Evaluation of crofelemer in the treatment of diarrhea-predominant irritable bowel syndrome patients. Digestion. 2008;78(4):180-6. doi: 10.1159/000185719. Epub 2008 Dec 18. PMID: 19092244

• Biswal S. Crofelemer: In HIV Associated Diarrhea and Secretory Diarrhea - A Patent Perspective. Recent Pat Antiinfect Drug Discov. 2014;9(2):136-43. doi: 10.2174/1574891x10666150408153356.PMID: 25851117

• Tradtrantip L, Namkung W, Verkman AS. Crofelemer, an antisecretory antidiarrheal proanthocyanidin oligomer extracted from Croton lechleri, targets two distinct intestinal chloride channels. Mol Pharmacol. 2010 Jan;77(1):69-78. doi: 10.1124/mol.109.061051. Epub 2009 Oct 6.

• Chordia P, MacArthur RD. Crofelemer, a novel agent for treatment of non-infectious diarrhea in HIV-infected persons. Expert Rev Gastroenterol Hepatol. 2013 Sep;7(7):591-600. doi: 10.1586/17474124.2013.832493.PMID: 24070150

• Nee J, Salley K, Ludwig AG, Sommers T, Ballou S, Takazawa E, Duehren S, Singh P, Iturrino J, Katon J, Lee HN, Rangan V, Lembo AJ. Randomized Clinical Trial: Crofelemer Treatment in Women With Diarrhea-Predominant Irritable Bowel Syndrome. Clin Transl Gastroenterol. 2019 Dec;10(12):e00110. doi: 10.14309/ctg.0000000000000110.PMID: 31800542

4. Does the journal accept previous publication in bioRxiv?

5. Control capsules or tablets? Varies at some places.

6. Definition of ≤7 stools as clinically significant for the definition-threshold of diarrhea could be emphasized. Is it specific for the animal model used?

7. Were the differences in stool consistency significant? The weekly mean fecal scores and stool consistency were 5.1, 3.9, and 4.1 for the CTR, BID, and QID groups (p<0.05). Might not have been a linear relationship.

8. Statistical methods show some variation; need to consistently follow non-parametric methods.

Reviewer #2: In this work, the authors perform a study to test the anti-diarrheal benefit of the medicine, crofelemer, in a beagle dog animal model over the course of 28 days. Specifically, the authors evaluated the efficacy of 125mg QID vs BID crofelemer prophylaxis would improve neratinib-induced diarrhea over QID placebo over the study period. The authors grouped animals into responders vs non-responders based off of the purina fecal scale of <6 representing a response with 6 and 7 considered loose or watery stools. The data was reported as average stool consistency for week 1, 4, or the entire 4 week period. In general, the authors determined that both crofelemer groups had improvement to phenotypic diarrhea without a significant difference between crofelemer groups. The study seeks to answer an important question of whether or not crofelemer should be added to the anti-diarrheal formulary for chemotherapy-induced diarrhea. The study lacks statistical rigor as raw data is batched into arbitrary categories of responder and non-responder, and statistics are largely performed on these categories rather than the raw data using PFS. Further, many data sets loosely referenced throughout the work (e.g., neratinib dose regimen changes, animals requiring subcutaneous fluids, episodes of emesis, bloodwork, and histopathology) are not provided. Recommend repeating statistics with non-batched data and including missing data sets. My comments exceed the word limit here, please see attached document.

6. PLOS authors have the option to publish the peer review history of their article (what does this mean?). If published, this will include your full peer review and any attached files.

Reviewer #1: No

Reviewer #2: No

---

## [Author Response · Author response to Decision Letter 0]

30 Jun 2023

Responses to reviewers from the authors are in green font throughout this document.

The authors are grateful for the time that was dedicated to a thorough review of this manuscript. All comments and suggestions were carefully addressed, and we believe the revised manuscript is vastly improved. Thank you.

Comments for the Authors:

• The study presents the results of primary scientific research.

o Yes

• Results reported have not been published elsewhere.

o Yes: This appears to be the same study that was published in BioRx in 2023 by the same authors, although that is not a formally peer-reviewed journal, so would leave it up to the PlosONE editor if this is ok to publish in this journal: “Guy, Michael K., et al. "Effects of orally administered crofelemer on the incidence and severity of neratinib-induced diarrhea in female dogs." bioRxiv (2023): 2023-02.”

Authors’ response: The reference created by BioRXiv is a “pre-print” reference for the current submission (please note the date “2023: 2023-02”). When the authors submitted this original manuscript to PlosONE, a reference was created and can be found online prior to having the manuscript fully accepted. For additional information please follow the link: https://journals.plos.org/plosone/s/preprints.

Thank you.

• Experiments, statistics, and other analyses are performed to a high technical standard and are described in sufficient detail.

o No. Only several brief and “representative” examples of the raw data are submitted in this work. The majority of statistics appear to have been performed using arbitrary cut offs to define responders vs non-responders rather than using the raw stool scale consistency numbers. And there were no data or statistics presented that describe multiple components of the study include hydration status, oral intake, dose changes, subQ fluid administration. 

Authors’ response: The authors are grateful for the reviewer’s concern. The responder analysis was defined a priori and was intended to be complementary to the study objective of directly measuring the number of loose/watery (non-formed) stools by treatment group that was presented in this manuscript. This has been further explained in the revised manuscript.

Please note that all four weeks of data collected during the study are now presented in the tables of results as well as comparisons between crofelemer BID and QID treatment groups.

Additionally, Supplementary tables have been included that contain information collected during the study about body weights, food consumption, volumes of subcutaneous administration, clinical adverse events, and clinical pathology data.

More information to follow in the “Specific Comments Section” below.

Thank you.

• Conclusions are presented in an appropriate fashion and are supported by the data. The main conclusion that crofelemer prophylaxis can reduce the incidence of diarrhea in this animal model could be supported by the data. But my concern is that the data presented is the result of several pre-statistical calculations using arbitrary cut-offs to batch label animals into responder vs non-responder phenotypes rather than using raw data for definitive analysis. The statistics here appear biased towards the hypothesized effect. 

Authors’ response: The revised manuscript more clearly presents the analysis of the numerical differences in the Purina Fecal Score (PFS) between treatment groups collected during the study. In addition, a responder dog was defined a priori as any dog that had ≤ 7 loose/watery stools per week (which averages out to be ≤ 1 loose/watery stool per day), for at least 2 of the 4 weeks of the 4-week study period.

More information to follow in the “Specific Comments Section” below.

Thank you.

• The article is presented in an intelligible fashion and is written in standard English.

o Yes

• The research meets all applicable standards for the ethics of experimentation and research integrity.

o Yes.

• The article adheres to appropriate reporting guidelines and community standards for data availability.

o No. Multiple data sets are absent and intermittently referenced in the discussion. These include data from weeks 2 and 3, data on food and water intake, episodes of emesis, which animals required changes in dosing regimen, including the size of the dose change and the timing of the change. There are no lab data sets reported even though the authors note that these were obtained. And the authors make several references to intestinal histology obtained after animals were sacrificed that was unremarkable and normal but these data were not shown.

Authors’ response: The authors believe the revised manuscript provides the reader with a more complete understanding of the data that was collected during this study. Weekly data from weeks 2 and 3 were added to several tables of results.

Supplementary tables are also included that contain information collected during the study about body weights, food consumption, volumes of subcutaneous administration, clinical side effects, and clinical pathology data.

More information to follow in the “Specific Comments Section”.

Thank you.

• Specific comments follow:

• Abstract

o Would include confidence intervals if there’s room

Authors’ response: The Abstract now includes the confidence intervals and p-values, and also comparisons between crofelemer BID and QID treatment groups.

Thank you.

o Why were these dosing regimens chosen? A similar study was performed in patients receiving trastuzumab, pertuzumab, and taxane in the HALT-D trial and crofelemer was administered as 125mg PO BID.

Authors’ response: Neratinib is known for its extreme diarrheagenic effects to the point where patients are directed on the Nerlynx Package Insert to start the use of anti-diarrheal opioids such as loperamide prior therapy initiation. Due to the high stool passage frequency reported in Nerlynx patients, the authors decided to investigate the possibility of both BID and QID crofelemer dosing regimens.

Thank you.

• Conflict of interest statement:

o The authors report no competing interests, however the lead author appears to be employed by Napo Pharmaceuticals which manufactures crofelemer under the trade name Mytesi ®, the drug being interrogated in this study. Unclear if Napo Pharmaceuticals funded this privately or not and this should be noted, as the authors reported that this project was not specifically funded. This should be noted.

Authors’ response: All of the authors have identified themselves as either Napo Pharmaceuticals employees, consultants or contractors. This study was performed to ethical standards and the authors reported their conflict of interest.

Thank you.

• Figure 1: Examples of non-responder and responder dogs

o Figure is difficult to interpret and appears low-quality, can this be converted to a graph? Are these respresentative of two single dogs? Why not show all the data?

Authors’ response: The authors agree with the reviewer and understand that this figure could be confusing. Since Figure 1 was an illustrative example and not representing any data, that Figure 1 has been eliminated from the revised manuscript.

Additionally, the authors created a paragraph that better describes the definition of a responder vs. non-responder dogs

Thank you.

• Figure 2: Percentage of dogs by treatment group that were determined to be responder dogs over the entire 4-week study period of crofelemer prophylaxis in neratinib-induced diarrhea in dogs.

o Recommend showing all data points on these graphs, would also define what is considered a responder in this graph. Further, categorizing animals as a responder vs non-responder despite still having loose and frequent stools is suggestive of some underlying bias, would consider graphing the raw data (e.g., graphs of loose stool number and stool consistency), also there are no statistics demonstrated on this graph, is this statistically significant?

Authors’ response: The authors have now inserted a revised Figure 1 to provide a clearer definition of responder dogs and the statistical differences in the proportion of responder dogs between treatment groups.

Additionally, a new paragraph was added to the Endpoints section of Materials and Methods that more fully describes the definitions of a responder dog and why this determination of a responder is so clinically important.

Authors’ note: In the revised manuscript, there are now only two figures:

Fig 1. Proportion of dogs that responded to crofelemer treatment in resolving neratinib-induced diarrhea by treatment group over the 4-week crofelemer study period in neratinib-induced diarrhea in dogs (n=8 per treatment group).

Supplementary Fig 1 (S1). Scoring chart for the Purina Fecal Score (PFS) System.

Thank you.

• Figure 3: Average number of watery stools by treatment group over the entire 4-week study period of crofelemer prophylaxis in neratinib-induced diarrhea in dogs.

o This graph lacks and needs statistics, are these not significantly different? Why is the graph changes to a horizontal rather than vertical? Would keep the order of the groups the same (figure 2 is CTR, QID, and then BID; while this graph is BID, QID, CTR)

Authors’ response: The authors agree with the reviewer that a table is more suited to report this data, including p-values and the contrast analysis between treatment groups and is presented in the revised manuscript as Table 2.

Thank you.

• Table 1: Average number of water stools per week by treatment group over the 4-week crofelemer study period in neratinib-induced diarrhea in dogs (n=8 per group).

o Would define “WS” in the figure legend, assuming it means “watery stools”, would also consider unifying the verbiage here since both WS and loose stools are used and appear to mean the same in this work, but if this is meant to convey a difference in severity, would state that more clearly; would keep group orders consistent between figures and tables; this table is useful for looking at the data, but a graph of this data should also be shown with ANOVA statistics comparing all groups, I’m assuming the p-values represent significant difference from CTR groups, but it is important to see whether or not there is a statistically significant difference between the experimental groups as well, which is not shown here, although clearly not significant

Authors’ response: The authors have also corrected the use of loose/watery stools throughout the manuscript for consistency.

Also, Table 1 now contains additional details including comparisons between crofelemer BID and QID treatment groups.

Thank you.

o Further, why were weeks 1 and 4 chosen, what happened in weeks 2 and 3?

Authors’ response: All tables now present data collected from all 4 weeks of the study, including weeks 2 and 3.

Thank you.

o I found the footnote helpful to understand this table, might include statistics down there as well

Authors’ response: Similar footnotes have been placed on the other tables in the revised manuscript.

Thank you.

• Table 2: Average number of loose/watery stools per week per neratinib dose by treatment group over the 4-week crofelemer study period in neratinib-induced diarrhea in dogs (n = 8 per group).

Authors’ note: Please note that Table 2 in the original manuscript is now Table 3 in the revised manuscript.

o Here “loose/watery” is used, would be consistent throughout the work.

Authors’ response: The authors have also corrected the use of loose/watery stools throughout the manuscript for consistency.

Thank you.

o I don’t feel that this figure adds too much, since all animals received the same regimen of neratinib, and only the first 5d used daily 40mg dosing and the following 23 days used 80mg once daily dosing, if there is a difference in stool output, would like to see some statistics comparing the loose stools per dose between the groups (consider a nested anova).

Authors’ response: Throughout the 28-day study there was non-significant variation in neratinib dosing between the three treatment groups, as presented in the revised manuscript as Table 2. 

The authors present this data in Table 2 to demonstrate that the changes in the number of loose/watery stools in Table 1 are not due to significant differences in the dosing of neratinib between the three treatment groups.

Thank you.

• Table 3: Average Purina fecal scores (PFS) for stool consistency per week by treatment group over the 4-week crofelemer study period in neratinib-induced diarrhea in dogs (n=8 per group).

Authors’ response: Due to the similarities between Table 1 and Table 3 in the original manuscript, Table 3 from the original manuscript was removed.

Author note: In the revised manuscript, there are now a total of 3 Results Tables and 5 Supplementary Tables:

Table 1. Average number of loose/watery stools per week by treatment group.

Table 2. Average daily number of neratinib tablets administered each week by treatment group.

Table 3. Average number of loose/watery stools per week per week per neratinib dose by treatment group.

S1 Table. Body weight at enrollment and weekly body weight changes from baseline by treatment group.

S2 Table. Food consumption per week by treatment group.

S3 Table. Weekly average volume (mL) of subcutaneous fluid administration each week by treatment group.

S4 Table. Adverse event profile from baseline to end of 4-week study period.

S5 Table. Number of dogs with out-of-range clinical chemistry and hematology parameters

o I am not familiar with the PFS system, would provide a key to accompany this table, a quick google search shows that it appears to be similar to the Bristol stool scale used in humans.

Authors’ response: Supplemental figure S1 was added with a description of the Purina Fecal Score (PFS) System that was used in this study. The PFS is the only fecal scoring system used in this study.

Thank you.

o Use consistent table order as recommended above.

Authors’ response: A consistent order in the tables with all weeks of data have been completed.

Thank you.

o Where is the data from Weeks 2 and 3?

Authors’ response: Results from weeks 2 and 3 are now included in the tables of Results.

Thank you.

o What is the difference between this table and table 1? This table seems to be more objective with use of the PFS scale.

Authors’ response: The authors agree with the reviewer and original Table 3 has been removed from the manuscript.

Thank you.

o Recommend showing statistics between all groups with nested anovas, are QID and BID significantly different from each other? I’m assuming not

Authors’ response: p-values showing comparisons between crofelemer BID and QID treatment groups have been added to the tables of results.

Thank you.

o Also, for all p-values in these tables, would make it more clear that they are comparing experimental to control groups

Authors’ response: The authors redesigned the tables to better display the various comparisons between treatment groups.

Thank you.

• Table 4: Trend of number of weekly dose reduction of neratinib by treatment group over the 4-week crofelemer study period in neratinib-induced diarrhea in dogs (n = 8 per group)

o Where is week 2-3 data?

Authors’ response: Results from weeks 2 and 3 are now included in the tables of Results.

Thank you.

o How were the doses reduced and what was the criteria for reduction?

Authors’ response: A new paragraph was created to clarify how dose reductions of neratinib were managed, the paragraph reads as follow: “Neratinib dose reductions or 1-day holidays were allowed when an individual study animal was deemed to be dehydrated, lethargic, or inappetent. Dose reductions were classified as dogs receiving 40mg of neratinib (1 tablet) instead of 80mg (2 tablets), while dose holidays allowed for 1-day only when animals were deemed incapable of receiving either 80mg or 40mg of neratinib. Data regarding neratinib tablets administration was collected and analyzed to avoid unbalanced dosages of neratinib between treatment groups.”

Thank you.

• Materials and Methods

o Animals

Why is randomization important here? Is this specifically due to differences in weight and age?

Authors’ response: Randomization was conducted to minimize individual variations between subjects, although all female beagle dogs were obtained from the same breeder/laboratory.

Body weight at enrollment was added to Supplemental Table 1 (S1 Table. Body weight at enrollment and body weight changes per week by treatment group.). 

Thank you.

o Study design

Why were these doses chosen? Are neratinib or/and crofelemer weight based? If so, these doses may lead to significantly higher drug concentrations than those used in patients undergoing treatment with neratinib or/and crofelemer since 125mg BID seems to be the current recommended dosing regimen for the 70kg human.

Authors’ response: The goal of this study was to create a sustainable neratinib-induced diarrhea model without life-threatening side effects to evaluate the potential anti-diarrheal effects of crofelemer.

Information from the pilot study, now included in the revised manuscript, determined an appropriate dose of neratinib that induced diarrhea in female beagle dogs but without causing life-threatening side effects.

Crofelemer tablets must be administered whole, without breaking or crushing. The NOAEL for crofelemer tablets in 12-week old beagle dogs is 156 mg BID, substantially higher than the single 125mg crofelemer tablet given BID or QID in this study. Administration of one 125mg crofelemer tablet given BID has also been investigated in a clinical study of dogs with acute diarrhea. It is also important to notice that crofelemer is minimally absorbed and acts locally in the G.I. tract.

Thank you. 

If these tablets are delayed-release, why is the QID regimen required?

Authors’ response: The authors chose to investigate QID dosing to potentially account for a more rapid physical washout of crofelemer due higher loose/watery stool passage frequency in cases of extreme diarrhea known to be present when dosing neratinib in humans.

Thank you.

What is the half-life of this formulation and average therapeutic window?

Authors’ response: Crofelemer is negilibly absorbed orally and acts from the luminla side of the gastrointestinal tract. Elctrophysiologicla studies (Trantrantip et al 2010) have reported a long pharmacodynamic half-life for crofelemer following experiments with CFTR ion channels. Clinical studies across various patiernt p[opulations indicate an average pharmacodymmic half-life of 12 hours for crofelemer effect. The half-life of crofelemer delayed-release tablets has not been evaluated in dogs. .

Thank you.

Further, it is noted in the table footnotes and in a later methods section that the dose of neratinib was increased to 80mg daily at day 6-23 due to emesis in the first few days, would include that in the study design portion for clarity.

Authors’ response: The authors agree that there were some inconsistencies when explaining the dose regimen for neratinib. The authors have revised the description of neratinib dosage for the 28-day study throughout the manuscript.

Thank you.

The sample size seems adequate to evaluate an effect against placebo, which confounders that were not controlled are being suggested here that are due to sample size?

Authors’ response: There were no confounders in this study.

Thank you.

o Housing and Husbandry

Is the 30-70% humidity range standard?

Authors’ response: Correct, this study followed Animal Welfare Act (USDA, 2013) regulations for indoor housing of laboratory animals. More was added to the Housing and Husbandry section of the revised manuscript. 

Thank you.

Were the intake volumes of food and water recorded and quantified by individual animals in this study? It would be good to know that all animals had approximately the same intake of food and water throughout this diarrhea study, especially since the neratinib dose is increased at day 6.

Authors’ response: The authors have included information about food and water consumption in the Results Section as well as in the Supplemental Material Section.

Thank you.

o Dose Administration, Animal Care, and Monitoring

It would be interesting to see the emesis data here, as well as a graph showing which doses were admitted from which animals for clarity; for example, did the dose or frequency of crofelemer affect emesis at all?

Authors’ response: Emesis data as we as other adverse events were added to the Supplemental Material Section.

Thank you.

What is meant by “neratinib was titrated daily”? in the study design, it is referenced as either 40mg or 80mg, this needs to be described and the data needs to be shown in better detail to improve the rigor of this work.

Authors’ response: The authors agree that there were some inconsistencies when explaining the dose regime for neratinib. The authors have revised the description of neratinib dosage for the 28-day study throughout the manuscript.

Thank you.

Further, there did not appear to be any tables or graphs comparing the symptoms of anorexia, lethargy, dehydration, vomiting, or bloody diarrhea, this should be included as part of therapeutic and adverse event monitoring

Authors’ response: The data collected regarding adverse events that were observed during the study is now presented in the Results Section as well as in the Supplemental Material Section.

Thank you.

o Data collection

Would include body weight, subQ fluid admin, and food data in this work rather than only referencing that it was recorded

Authors’ response: The authors have included information about subcutaneous fluid administration in the Results Section as well as in the Supplemental Material Section.

Thank you.

Fecal score was collected twice daily in this study, is this an aggregate of several bowel movements? Would consider including the mass or volume of the stools in these collections as part of the quantitative data. How do the animals defecate? Through the cages, is there a designated area?

Authors’ response: Fecal Score was collected twice daily prior to the cage floor cleaning (through passage to metal tray). The fecal samples were not collected or weighed.

Thank you.

o Ethical Statement

It is noted that blood was drawn for clinical pathology but this data was not presented, would include if possible, as well as any tissue data that was collected (for example, any evaluation of intestinal epithelium between the groups?)

Authors’ response: Supplementary Table S5 contains a summary of significant clinical pathology results noted during the 4-week study.

Thank you.

Would consider referencing standard euthanasia protocol here.

Authors’ response: The standard euthanasia protocol used in this study is part of the Ethical Statement found within the Methods and Materials section.

Thank you.

o Statistical Methods

It is noted that no data was excluded, but data from Weeks 2 and 3 are generally missing throughout this work, would include these

Authors’ response: Results from weeks 2 and 3 are now included in the tables of Results.

Thank you.

It is noted that PFS of 6 or 7 was considered watery bowel movements and anything <6 was considered normal. However, according to the published PFS, only 7 should be considered watery, 6 would be considered “has texture, but not defined shape; occurs as piles or as spots; leaves residue when picked up”; and would use caution with calling 1 or 3-5 as “normal” as these are not described as normal in this published literature. Would consider not grouping PFS numbers together and simply quantify the raw data. This could be done in a scatter plot format, with each experimental group represented by a different line, similar to what has been shown for murine norovirus (copied here, “Roth, Alexa N., et al. "Norovirus infection causes acute self-resolving diarrhea in wild-type neonatal mice." Nature Communications 11.1 (2020): 2968.”):

Authors’ response: The authors have revised the manuscript to be more consistent when defining the dichotomization of the data.

Thank you.

o I would caution against displaying this data as responder vs non-responder and would base an effect more on the stool consistency as an objective measure

Why was <7 watery bowel movements per week chosen for the responders? What is the average normal expected watery bowel movement for this animal model?

Authors’ response: The “responder analysis” was defined a priori as any dog that had ≤ 7 loose/watery stools per week (which averages out to be ≤ 1 loose/watery stool per day), for at least 2 of the 4 weeks of the 4-week study period. The responder analysis is intended to be complementary to the study objective of directly measuring the number of loose/watery stools by treatment group that was presented in this manuscript.

More was added to the body of the revised manuscript.

Thank you.

How was the dose of neratinib required to maintain a consistent level of diarrhea? Was this determined in the study or was there a preliminary study used to determine this? If the dose of neratinib had to be altered to maintain a diarrheal phenotype in the placebo, than that would argue against the utility of crofelemer during neratinib treatment since animals receiving placebo adapted to neratinib w/o specific intervention. It is not enough to say that subQ fluid admin and dose changes of neratinib were analyzed using similar procedures, since these two aspects alone could entirely explain any alterations in fluid-content of stools in this study. This data needs to be shown, explained, and statistics analyzed in a dedicated figure or table.

Authors’ response: Information from the pilot study (now included in the revised manuscript) determined an appropriate dose of neratinib that induced diarrhea in female beagle dogs but without causing life-threatening side effects.

Supplementary tables are also included that contain information collected during the study about body weights, food consumption, volumes of subcutaneous administration, clinical side effects, and clinical pathology data. The results from these variables showed no significant differences. The authors agree that if those variables had major differences between the groups, a more refined analysis would have to be conducted.

Thank you.

Is the effect size in this study 1.0? If so, the study may be underpowered, but that does appear to be noted elsewhere.

Authors’ response: Effect sizes for all the efficacy endpoints have been computed and are listed in the supplemental tables; they all were approximately 1 for the key efficacy endpoints. Since many of the efficacy endpoints yielded statistically significant differences between crofelemer groups and control, the sample size was adequate, and the study was not underpowered.

Thank you.

o Results (reviewer received a non-numbered article)

First sentence: authors describe there were no deleterious effects on the dog’s behavior, however in the methods they noted that some animals had emesis requiring modifications to the study, would describe that more consistently

Authors’ response: Emesis data as we as other adverse events were added to the Supplemental Material Section.

Thank you.

I like that the animal and human dose comparison was made, would include in the methods section, however a quick google search shows that common starting regimens of neratinib is 240mg once daily for a year which is ~3.4mg/kg for a 70kg human, would clarify. There is also some reference to even less in the starting dose, around 120mg daily with a slow up-titration.

Authors’ response: The authors hope that the previous responses about dosing have addressed this concern.

Thank you.

See figure comments for specific comments related to the data

Authors’ response: 

Thank you.

Would simply describe the raw data using the stool scale numbers, my concern is that the effect of responder vs non-responder may be overstating the effect when a more careful analysis of the PFS numbers would reveal more subtle changes in stool consistency that would be important points to make when translating this work to patients who are often on multiple anti-diarrheals. For example, in clinic, we don’t say “responder” or “non-responder”, we like to quantify the stool number and consistency, and characterize the stool as bloody vs non-bloody, along with other symptoms such as abdominal pain, etc. Inclusion of this data would improve the rigor of this work.

Authors’ response: The revised manuscript more clearly presents the analysis of the numerical differences in the Purina Fecal Score (PFS) between treatment groups collected during the study. In addition, a responder dog was defined a priori as any dog that had ≤ 7 loose/watery stools per week (which averages out to be ≤ 1 loose/watery stool per day), for at least 2 of the 4 weeks of the 4-week study period. Thank you.

Week 2 and 3 data is not mentioned aside from overall data, is there an expectation for delayed response to crofelemer?

Authors’ response: Results from weeks 2 and 3 are now included in the tables of Results.

Thank you.

Would mention if there is a statistically significant difference between treatment groups, rather than only compared to placebo group.

Authors’ response: The p-values comparing the crofelemer BID and QID groups are now presented in all tables of results

Thank you.

Paragraph that references table 3: the first sentence doesn’t make sense, would check for accuracy. What is meant by “significantly greater stools”? I like that the PFS are reported here, would also show daily trends per group. It is also important to note any changes to dosing regimens made to maintain diarrheal phenotype as briefly described in the methods. Some grammar corrections need to be made.

Authors’ response: Thank you for the suggestion. The passage was revised.

Thank you.

The data in table 4 does not appear to be statistically significant in the current format, would consider showing the individual animal data points here rather than averages. Table 4 does not describe subQ fluids, would include that data.

Author note: In the revised manuscript, there are now 3 Results Tables, and 5 Supplementary Tables:

Table 1. Average number of loose/watery stools per week by treatment group.

Table 2. Average daily number of neratinib tablets administered each week by treatment group.

Table 3. Average number of loose/watery stools per week per week per neratinib dose by treatment group.

S1 Table. Body weight at enrollment and weekly body weight changes from baseline by treatment group.

S2 Table. Food consumption per week by treatment group.

S3 Table. Weekly average volume (mL) of subcutaneous fluid administration each week by treatment group.

S4 Table. Adverse event profile from baseline to end of 4-week study period.

S5 Table. Number of dogs with out-of-range clinical chemistry and hematology parameters

.

I like that the path was described in the final section, but would include the H&E of the distal ileum and colonic mucosa at the very least as a figure as well. There is also evidence in the literature for histopathologic changes in distal ileal and colonic mucosa following 2-4 weeks of neratinib in other animal models, would contrast your findings with those of prior work and justify why the results are different (different animal model vs dosing regimen, etc).

Authors’ response: The authors regret the electronic images of histopathology slides from this study are not available.

Thank you.

o Introduction

References are inconsistent in the sentences, would fix in final copy-editing

Authors’ response: References have been corrected.

Thank you.

I like this introductory paragraph, but would add a sentence of the mechanism by which TKIs lead to diarrhea going beyond simply stating that they target EGFR, TKs, and HER2--are there changes in ion transport, inflammation with decreased intestinal barrier integrity? Etc

Authors’ response : The inhibition of EGFRs by TKIs and other targeted therapies does not alter intestinal barrier integrity. However, due to the inflammation induced by targeted therapies from inhibition of EGFR targets, intestinal chloride ion secretion is increased by inhibition of EGFRs, as EGFRs is a negative regulator of intestinal chloride ion channels. 

Would add reference for function of crofelemer, also an earlier description of the ion-transport basis of diarrhea would clarify why the authors chose this chloride-ion transporter inhibitor to study

Authors’ response: The function and MOA of crofelemer are more fully described in the Introduction.

Thank you.

What is the actual mechanism by which crofelemer inhibits Cl- transport? Does it sterically block these channels?

Authors’ response: Crofelemer is an inhibitory chloride ion channel modulator of both CFTR and/or CaCC. More was added to the Introduction to address the MOA of crofelemer on chloride ion regulation.

Thank you

o Discussion

Nice background on mechanism underlying crofelemer, would put this in the intro.

Authors’ response: This paragraph has been moved to the Introduction.

Thank you.

Recommend showing the data with regards to body weight, hydration status, food consumption, and “other hematological or clinical chemistry parameters”

Authors’ response: Supplementary tables are included that contain information collected during the study about body weights, food consumption, volumes of subcutaneous administration, clinical side effects, and clinical pathology data.

Thank you.

It is notable that the authors state the absence of histopathologic changes in intestinal tissues was consistent to the findings of that in a human study (Liu et al 2022), when in that study, only ~7% of patients receiving TKI therapy underwent colonoscopies, with ~50% showing no path, ~33% with non-ulcerative inflammation, and the rest with ulcers. Thus the results in this work, which are only stated and not shown, are not consistent with this prior study.

Authors’ response: This paragraph has been revised.

Thank you.

New data is being presented in the discussion with a 6% neratinib dose increase in crofelemer treated animals and a %% reduction in subQ fluid replacement. New data should not be presented in the discussion, and this data should have adequate graphical or tabular descriptions with appropriate statistical analysis. Further, 6% dose increases were not noted in the study design, it is unclear why this was done.

Authors’ response: These discrepancies have been addressed in the Results section to better explain the differences between treatment based on dose reduction and holidays and the difference between treatment. No new data is presented in the Discussion.

Thank you.

Throughout the discussion and this work, multiple different stool consistency grading mechanisms are introduced without description or an example chart, would add either a description or a chart for understanding.

Authors’ response: Supplemental figure 1 was added with a description of the Purina Fecal Score (PFS) System that was used in this study. The PFS is the only fecal scoring system used in this study.

Thank you.

The loperamide trials in the discussion would be more appropriate in the introduction as a reason why we need new and improved anti-diarrheals, but definitely an important point to make.

Authors’ response: Great suggestion. The authors will move this paragraph to the introduction.

Thank you.

Authors again reference that there are no histopathologic findings in prior studies when there actually were both macroscopic and microscopic findings in citations 14, 16, and 17. Of note, citation 17 is a case report of a patient with UC, status-post colectomy, who presented with duodenitis that was successfully treated with tacrolimus and does not appear to be relevant to this work.

Authors’ response: This paragraph has been revised.

Thank you.

The crofelemer background is repeated in the discussion, can take this out, but would also describe the half-life and bioavailability in more detail given the frequent (QID) dosing regimen used in this study. This could explain the lack of difference between BID and QID groups.

Authors’ response: Neratinib is known for its extreme diarrheagenic effects to the point where patients are directed on the Nerlynx Package Insert to start the use of anti-diarrheal opioids such as loperamide prior therapy initiation. Due to the high stool passage frequency reported I Nerlynx patients, the authors decided to investigate the possibility of QID dosing as an additional treatment group. Also, the half-life of this preparation of crofelemer is unknown in dogs with neratinib-induced diarrhea.

Thank you.

o Acknowledgements

The authors report that there are no financial or non-financial conflicts of interest as all authors are employees or consultants of Napo Pharmaceuticals, Inc. Which is actually a notable COI, because it is this company that markets crofelemer.

Authors’ response: The conflict of interest was addressed and will be appropriately described during the submission of the reviewed manuscript.

Thank you.

---

## [Decision Letter · Decision Letter 1]

10 Oct 2023

PONE-D-23-05162R1Effects of orally administered crofelemer on the incidence and severity of neratinib-induced diarrhea in female dogsPLOS ONE

Dear Dr. Guy,

Thank you for submitting your manuscript to PLOS ONE. After careful consideration, we feel that it has merit but does not fully meet PLOS ONE’s publication criteria as it currently stands. Therefore, we invite you to submit a revised version of the manuscript that addresses the points raised during the review process.

At this point the concerns of the reviewer 2 are relatively minor.  Please respond to this comments and revise the manuscript accordingly. Please submit your revised manuscript by Nov 24 2023 11:59PM. If you will need more time than this to complete your revisions, please reply to this message or contact the journal office at plosone@plos.org. Please include the following items when submitting your revised manuscript:A rebuttal letter that responds to each point raised by the academic editor and reviewer(s). You should upload this letter as a separate file labeled 'Response to Reviewers'.A marked-up copy of your manuscript that highlights changes made to the original version. You should upload this as a separate file labeled 'Revised Manuscript with Track Changes'.An unmarked version of your revised paper without tracked changes. You should upload this as a separate file labeled 'Manuscript'.If applicable, we recommend that you deposit your laboratory protocols in protocols.io to enhance the reproducibility of your results. Protocols.io assigns your protocol its own identifier (DOI) so that it can be cited independently in the future. For instructions see: https://journals.plos.org/plosone/s/submission-guidelines#loc-laboratory-protocols. Additionally, PLOS ONE offers an option for publishing peer-reviewed Lab Protocol articles, which describe protocols hosted on protocols.io. Read more information on sharing protocols at https://plos.org/protocols?utm_medium=editorial-email&utm_source=authorletters&utm_campaign=protocols.

We look forward to receiving your revised manuscript.

Kind regards,

Pradeep Dudeja

Academic Editor

PLOS ONE

Journal Requirements:

Reviewers' comments:

Reviewer's Responses to Questions

**Comments to the Author**

1. If the authors have adequately addressed your comments raised in a previous round of review and you feel that this manuscript is now acceptable for publication, you may indicate that here to bypass the “Comments to the Author” section, enter your conflict of interest statement in the “Confidential to Editor” section, and submit your "Accept" recommendation.

Reviewer #1: All comments have been addressed

Reviewer #3: (No Response)

2. Is the manuscript technically sound, and do the data support the conclusions?

Reviewer #1: Yes

Reviewer #3: Yes

3. Has the statistical analysis been performed appropriately and rigorously? 

Reviewer #1: Yes

Reviewer #3: Yes

4. Have the authors made all data underlying the findings in their manuscript fully available?

Reviewer #1: Yes

Reviewer #3: Yes

5. Is the manuscript presented in an intelligible fashion and written in standard English?

Reviewer #1: Yes

Reviewer #3: Yes

6. Review Comments to the Author

Reviewer #1: The authors have provided a reasonable improvement in the original manuscript. It will add to the available information on the effects of crofelemer on neratinib-induced diarrhea in female dogs. I am recommending accept.

Reviewer #3: Most of the concerns raised by the reviewers have been duly addressed.

However, there are some minor concerns:

1. As requested by the reviewers, the authors failed to include mass or volume of stools in the fecal score collections (as part of the quantitative data). Alternatively, the authors can provide data with regards to colon weight/colon length as a measure of diarrhea

2. Why only female beagle dogs were considered for the current study?

3. Supplemental Tables: 3 (average volume of subcutaneous fluid administration each week by treatment group and 4 (side effect profile throughout the 4-week study across treatment groups) are missing

7. PLOS authors have the option to publish the peer review history of their article (what does this mean?). If published, this will include your full peer review and any attached files.

Reviewer #1: No

Reviewer #3: No

---

## [Author Response · Author response to Decision Letter 1]

16 Oct 2023

Responses to reviewers from the authors were prepared on October 16th, 2023.

As requested, the Supplemental Figures and Tables have been removed from the manuscript and have been submitted in a separate file titled “Supporting Information’. 

Also, the legend for each item in the ‘Supporting Information’ file has been listed in the manuscript after the references list and is listed below:

Supplemental Information

Supplemental Figure S1. Scoring chart for the Purina Fecal Scoring (PFS) System

Supplemental Table S1. Body weight at enrollment and body weight changes per week by treatment group over the 4-week crofelemer study period in neratinib-induced diarrhea in dogs (n=8 per treatment group). 

Supplemental Table S2. Food consumption per week by treatment group over the 4-week crofelemer study period in neratinib-induced diarrhea in dogs (n=8 per treatment group).

Supplemental Table S3. Weekly average volume (mL) of subcutaneous fluid administration each week by treatment group over the 4-week crofelemer study period in neratinib-induced diarrhea in dogs (n=8 per treatment group). 

Supplemental Table S4. Adverse event profile from Days 0 through 28 of 4-week crofelemer study period in neratinib-induced diarrhea in dogs (n=8 per treatment group.

Supplemental Table S5. Number of dogs with out of range clinical chemistry and hematology parameters on study Days 0, 7, 14, 21, and 28 by treatment group over the 4-week crofelemer study period in neratinib-induced diarrhea in dogs (n=8 per treatment group).

---

## [Decision Letter · Decision Letter 2]

13 Nov 2023

Effects of orally administered crofelemer on the incidence and severity of neratinib-induced diarrhea in female dogs

PONE-D-23-05162R2

Dear Dr. Guy,

We’re pleased to inform you that your manuscript has been judged scientifically suitable for publication and will be formally accepted for publication once it meets all outstanding technical requirements.

Kind regards,

Pradeep Dudeja

Academic Editor

PLOS ONE

Additional Editor Comments (optional):

Reviewers' comments:

Reviewer's Responses to Questions

**Comments to the Author**

1. If the authors have adequately addressed your comments raised in a previous round of review and you feel that this manuscript is now acceptable for publication, you may indicate that here to bypass the “Comments to the Author” section, enter your conflict of interest statement in the “Confidential to Editor” section, and submit your "Accept" recommendation.

Reviewer #3: All comments have been addressed

2. Is the manuscript technically sound, and do the data support the conclusions?

Reviewer #3: Yes

3. Has the statistical analysis been performed appropriately and rigorously? 

Reviewer #3: Yes

4. Have the authors made all data underlying the findings in their manuscript fully available?

Reviewer #3: Yes

5. Is the manuscript presented in an intelligible fashion and written in standard English?

Reviewer #3: Yes

6. Review Comments to the Author

Reviewer #3: (No Response)

7. PLOS authors have the option to publish the peer review history of their article (what does this mean?). If published, this will include your full peer review and any attached files.

Reviewer #3: No

---

## [Editor Report · Acceptance letter]

2 Jan 2024

PONE-D-23-05162R2 

PLOS ONE

Dear Dr. Guy, 

I'm pleased to inform you that your manuscript has been deemed suitable for publication in PLOS ONE. Congratulations! Your manuscript is now being handed over to our production team.

Kind regards, 

on behalf of

Dr. Pradeep Dudeja 

Academic Editor

PLOS ONE